# Amazonia Phytotherapy Reduces Ischemia and Reperfusion Injury in the Kidneys

**DOI:** 10.3390/cells12131688

**Published:** 2023-06-22

**Authors:** Brenner Kássio Ferreira de Oliveira, Eloiza de Oliveira Silva, Sara Ventura, Guilherme Henrique Ferreira Vieira, Carla Djamila de Pina Victoria, Rildo Aparecido Volpini, Maria de Fátima Fernandes Vattimo

**Affiliations:** 1School of Nursing, University of São Paulo, São Paulo 05403-000, Brazil; eloizaosilva@usp.br (E.d.O.S.); sara.ventura@usp.br (S.V.); guihenrique@usp.br (G.H.F.V.); carlavictoria2001@usp.br (C.D.d.P.V.); nephron@usp.br (M.d.F.F.V.); 2Faculty of Medicine, University of São Paulo, São Paulo 01246-903, Brazil; rildo.volpini@hc.fm.usp.br

**Keywords:** acute kidney disease, phytotherapy, ischemia/reperfusion

## Abstract

Acute kidney injury (AKI) is defined as a sudden decrease in kidney function. Phytomedicines have shown positive effects in the treatment of AKI worldwide. The aim of this study was to evaluate the effect of *Abuta grandifolia* on the renal function of rats submitted to AKI. A phytochemical study of the plant was performed through liquid chromatography coupled with mass spectrometry (CL-EM) and DPPH and ABTS antioxidant tests. Renal function tests were performed in 20 male adult Wistar rats weighing from 250 to 300 g distributed in the following groups: SHAM (submitted to laparotomy with simulation of renal ischemia); ABUTA (animals that received 400 mg/kg of AG, orally—VO, once a day, for 5 days, with simulation of renal ischemia); I/N (animals submitted to laparotomy for clamping of bilateral renal pedicles for 30 min, followed by reperfusion); ABUTA + I/R (animals that received AG—400 mg/kg, 1× per day, VO, for 5 days, submitted to renal ischemia after treatment with herbal medicine). The results suggest that the consumption of *Abuta grandifolia* promoted renoprotection, preventing the reduction of renal function induced by ischemia, oxidizing activity, and deleterious effects on the renal tissue, confirmed by the decrease of oxidative metabolites and increase of antioxidants in the animals’ organisms.

## 1. Introduction

Acute kidney injury (AKI) describes an acute deterioration of renal function without total loss of function. It is a common and serious complication in patients in intensive care units with a significant impact on patient mortality and morbidity [1]. It affects 10% to15% of all hospitalized patients and approximately one in three intensive care unit (ICU) patients, and its incidence has increased from 35.8% to 50.4% [2,3].

Intensive care unit acute kidney injury (iAKI) is an independent risk factor for death, with mortality rates ranging from 40% to 55%. In addition to mortality, iAKI survivors are more likely to develop significant morbidity, including chronic kidney disease, end-stage kidney disease, and functional impairment that requires discharge to short-term or long-term care institutions [4].

Regarding the causes of AKI, it is thought they include any clinical conditions that result in interruption of renal blood flow, such as intravascular volume depletion, reduction of cardiac output, and vasodilator or vasoconstrictor disorders [5]. iAKI is associated with changes in hemodynamics and dysfunction of endothelial cells due to high levels of reactive oxygen species (ROS) and reactive nitrogen species (RNS), leading to decreased nitrogen oxide production and exhaustion of intracellular energy reserves [6].

Both ROS stress and RNS cause lipid peroxidation, oxidative DNA damage, modification of inflammatory pathways, modification of leukocyte function, and microvascular reduction in blood flow to the renal medulla due to vascular congestion. ROS is involved in kidney injury by lipid peroxidation, while oxidative damage of proteins and DNA contributes to apoptosis and cell necrosis [7]. 

Given its high prevalence and mortality, despite continuous and substantial improvement in intensive care and dialysis techniques, improving iAKI outcomes is a challenge [8]. New therapies should be investigated, such as those that explore traditional medicine, including the use of medicinal plants [9]. In the Amazon region, the richness of biodiversity is a fact, however, there is no scientific evidence on the safe and effective medicinal use of native plants [10].

Among the medicinal plants used, we highlight *Abuta grandifolia* (Mart.) Sandwith (Menispermaceae), an Amazonian plant that is popularly used for the treatment of malaria, diabetes, inflammation, rheumatism, hemorrhages, gynecological diseases, and kidney diseases. Regarding its phytochemical profile, it is known so far that there is the presence of secondary metabolites, such as alkaloid groups, flavonoids, saponins and tannins. However, there is still no scientific basis for the effects of these substances on kidney function [11]. 

In the case of iAKI, the hypothesis of this study is that *Abuta grandifolia*, already used in a traditional empirical way, may demonstrate a renoprotective impact, especially due to its antioxidant and anti-inflammatory properties. In addition, because several aspects of this species are not yet well elucidated, additional benefits can be identified for the control of this severe syndrome. Therefore, the hypothesis of this study is that *Abuta grandifolia* may exert a renoprotective effect in the situation of acute insult due to ischemia. The aim of this study is to evaluate the effect of *Abuta grandifolia* on renal function, hemodynamics, oxidative profile and histological analysis of the kidneys of rats submitted to iAKI.

## 2. Materials and Methods

The stems and leaves of *Abuta grandifolia* were harvested in the rural area of the Municipality of Parintins, Amazonas, Brazil. The exsiccata of the plant was identified and deposited in the Herbarium of the Federal Institute of Education, Science and Technology of Amazonas, under proof number 18433. The stems were dried in an oven at a temperature of 60 °C for 7 days, then crushed in an electric knife mill, and the resulting coarse powder was refined in a blender [12]. 

Regarding the particular compounds, the aqueous extract (AE) was analyzed by liquid chromatography coupled with a mass spectrometry detector (CL-MS) according to standard procedure [13,14,15]. Evaluation of the antioxidant activity of the aqueous sample was verified through DPPH and ABTS tests [16,17]. 

The determination of the sample size was analyzed using G*Power software, using a one-way ANOVA unidirectional variance test (F-test) [18]. The subjects were 20 Wistar rats, males, adults, 60 days old, weighing from 250 to 300 g. They were kept with free access to water and feed, in adequate thermal conditions and in alternating cycles of day and night. They were randomly allocated to four (4) groups: SHAM (*n* = 5), animals submitted to laparotomy to simulate clamping of bilateral renal pedicles for 30 min; ABUTA (*n* = 5), animals that received *Abuta grandifolia* (400 mg/kg, 1× daily, orally—VO, for 5 days, with simulation of renal ischemia); Ischemia (I/R) (*n* = 5), animals submitted to laparotomy for clamping of the bilateral renal pedicles for 30 min, followed by reperfusion; ABUTA + I/R (*n* = 5), animals that received *Abuta grandifolia* (400 mg/kg, 1× daily, orally—VO, for 5 days), submitted to renal ischemia after treatment with herbal medicine.

The rats were anesthetized with ketamine and xylazine (100 mg/kg and 10 mg/kg, respectively) and submitted to laparotomy for bilateral clamping of the renal pedicles for 30 min, with non-traumatic vascular forceps. All animals were evaluated during anesthetic recovery. 

After 24 h of rest for post-anesthetic recovery, the animals were placed in metabolic cages for 24-h urine collection to study renal function and oxidative stress. All animals were evaluated during anesthetic recovery and received post-procedure analgesic (Tramadol 15 mg/kg Intramuscular 3× a day for 2 days). After this period, the rats were removed from the cages, anesthetized, and submitted to procedures for the analysis of renal function by the technique of determining the inulin clearance.

The animals were submitted to laparotomy, and terminal blood collection by abdominal aortic puncture. The right kidney was removed for weighing and for purposes of calculation related to the animal’s weight/kidney, and the left kidney was removed for conditioning in a refrigerator at −80 °C to verify non-protein thiols. At the end of the experiment, the researcher euthanized the animals by anesthetic supplementation and collection of terminal blood, according to the ethical norms for handling animals in the laboratory [19].

The glomerular filtration rate (GFR) was determined by the inulin clearance technique. Preoperatively, the animals received analgesia with morphine (3 mg/kg) and were anesthetized with ketamine and xylazine (100 mg/kg and 10 mg/kg, respectively). After they were anesthetized, the operation began with the catheterization of the animal’s jugular vein for infusion of inulin in a bolus of initial dose of 100 mg/kg of body weight of diluted inulin, followed by continuous infusion of 10 mg/kg of body weight for the 2 h of the experiment, at a speed of 0.04 mL/min. Urine collection was performed every 30 min through bladder catheterization, and blood samples every 60 min. The sedation of the animals was maintained through an additional dose of xylazine, if necessary. The quantification of inulin concentration in the samples was performed by the Anthrone colorimetric method [20,21]. 

The serum creatine dosage was verified by the Jaffe colorimetric method [22]. 

Renal hemodynamics were verified through renal blood flow (RBF) with an ultrasonic probe (T402; Transonic Systems, Bethesda, MD, USA), which involved the left renal artery. Renal vascular resistance (RVR) was calculated using the following formula: RVR = MAP/RBF Measurement of the mean arterial pressure (MAP) was performed by catheterization of the carotid artery for catheter insertion (polyethylene tube —PE 60), and terminal blood was collected by puncture of the abdominal aorta, and the kidneys were sectioned for histopathological studies [23,24].

The evaluation of the oxidative profile occurred by measuring urinary peroxides (UP by the FOX-2 method [25]. The final products of lipid peroxidation were detected by the TBARS method (urinary thiobarbituric acid reactive substances) [26]. Nitrous oxide (NO) was measured by the Griess method [27]. Non-protein soluble thiols in the renal tissue with correction of the total proteins was quantified by the Ellman method [28].

In renal histology to assess tubulointerstitial involvement, a score graduated on a scale of 0 to 4 [20] was used, where 0 = normal; 0.5 = small focal areas; 1 = involvement of less than 10% of the cortex and external renal medulla, 2 = involvement of 10% to 25% of the external renal cortex; 3 = involvement of 25% to 75% of the cortex and external renal medulla; and 4 = extensive changes of more than 75% of the cortex and external medulla. The images obtained by optical microscopy were captured by means of a light video camera connected to an image analyzer and analyzed by fields of 0.245 mm^2^ of slides containing a sample of the renal tissue of each animal.

Data analysis was performed using one-way ANOVA. A significance level of *p* < 0.05 was chosen, considering that the effect of at least one of the groups was different from the others. Tukey’s 2 to 2 multiple comparison tests were used to assess which groups differed from each other.

## 3. Results

### 3.1. Phytochemical Analysis

The chromatographic profile by CL-MS of the aqueous extract (EA) provided relevant information about its possible phytochemical composition. The AS screening was performed with detection at 350 nm. Based on the key fragmentations observed and comparisons with mass/load (*m*/*z*) data found in the literature, the presence of the following secondary metabolites in the aqueous extract of *Abuta grandifolia* was inferred: the alkaloids nornuciferin [29], *N*-methyl-O-methylisopyrin [30], *N*-methyllaurothetanine, laurotetannin [31], menisperine, magnoflorin [32,33], erythramine and erythrascine [34], *N*-trans-feruloylthyramine [35], jatrorizine and palmatin [36], the flavonoid epicatechin [37], and finally, oleanolic acid [38]. All have anti-inflammatory and antioxidant action, reported in the literature. 

Antioxidant tests (Table 1) were performed in triplicate. The DPPH test showed that *Abuta grandifolia* has antioxidant properties of 76.03%, while for the ABTS test it presented antioxidant activity of 89.55%. As a result, there was inhibition of DPPH and ABTS may be related to the existence of alkaloids and flavonoids found in plants, which act as efficient antioxidants.

### 3.2. Renal Function Analysis

In the results presented below, the animals in the SHAM group were used as healthy controls and the data obtained for this group were considered parameters for normality.

### 3.3. Physiological Parameters

The weight of the animals and their kidneys and the kidney weight/weight ratios of the animals were measured only at the end of the experiment. As shown in Figure 1, there was a significant increase in the kidney weight/weight ratio of the animal in the ABUTA + I/R group, when compared to the SHAM, ABUTA and I/R groups (ABUTA + I/R: 0.50 ± 0.09 vs. SHAM: 0.32 ± 0.04; ABUTA: 0.40 ± 0.05 and I/R: 0.42 ± 0.06, *p* < 0.0001).

### 3.4. Renal Function

Renal function was assessed by measuring urinary flow, serum creatinine, and glomerular filtration rate, which was estimated by inulin clearance. Figure 2 shows the results for renal function. The ABUTA + I/R group presented a significant increase in urinary flow when compared to the other groups (ABUTA + I/R: 0.025 ± 0.004 vs. SHAM: 0.010 ± 0.001; ABUTA: 0.013 ± 0.003; I/R: 0.011 ± 0.005, *p* < 0.0001).

In the evaluation of serum creatinine, the I/R group showed an increase compared to the SHAM and ABUTAE groups (I/R: 2.30 ± 0.63 vs. SHAM: 0.30 ± 0.05; ABUTA: 0.30 ± 0.10, *p* < 0.0001), and the ABUTA + I/R group showed a decrease in creatinine indices in relation to the I/R group (ABUTA + I/R:0.63 ± 0.22 vs. I/R: 2.30 ± 0.63, *p* < 0.0001).

### 3.5. Renal Hemodynamics

Renal hemodynamics were evaluated by checking heart rate (HR), mean arterial pressure (MAP), renal blood flow (RMF) and renal vascular resistance (RVR). The ABUTA group showed a significant increase in the heart rate parameter when compared to SHAM (ABUTA: 584 ± 6 vs. SHAM: 492 ± 47, *p* < 0.0001), as shown in Figure 3.

In the MAP parameter, the groups did not present significant differences. Regarding RVR, there was no statistically significant difference between the SHAM and ABUTA groups. However, the I/R group showed increased RVR in relation to the SHAM and ABUTA groups (I/R: 56 ± 8 vs. SHAM: 11 ± 2; ABUTA: 11 ± 1, *p* < 0.0001). Treatment with *Abuta* significantly reduced RVR in the ABUTA + I/R group compared to the I/R group (ABUTA + I/R: 18 ± 9 vs. I/R: 56 ± 8, *p* < 0.0001). 

The I/R group showed a significant reduction in FSR in relation to the SHAM and ABUTAE groups (I/R: 2.0 ± 0.3 vs. SHAM: 9.2 ± 1.7; ABUTA: 7.2 ± 0.7, *p* < 0.0001), and ABUTA + I/R showed a significant increase in FSR in relation to I/R (ABUTA + I/R: 4.0 ± 1.7 vs. I/R: 2.0 ± 0.3, *p* < 0.0001).

### 3.6. Oxidative Profile

Figure 4 shows the results of the oxidative profile. In the urinary peroxide parameter (FOX), the I/R group showed a significant increase when compared to the SHAM and ABUTA groups (I/R: 16.2 ± 2.2 vs. SHAM: 3.3 ± 0.6; ABUTA: 6.0 ± 0.4, *p* < 0.0001), and the ABUTA + I/R group showed a reduction in this parameter in relation to the I/R group (ABUTA + I/R: 10.4 ± 3.5 vs. I/R: 16.2 ± 2.2, *p* < 0.0001).

In the lipid peroxidation parameter (TBARS), the I/R group showed a significant increase in relation to the SHAM group (I/R:1.90 ± 0.15 vs. SHAM: 0.20 ± 0.01, *p* < 0.0001). The ABUTA + I/R group showed a decrease in relation to the I/R group (ABUTA + I/R: 0.41 ± 0.06 vs. I/R: 1.90 ± 0.15, *p* < 0.0001).

For measurement of nitric oxide (NO), the I/R group showed an increase in relation to the SHAM and ABUTA groups (I/R: 174 ± 13 vs. SHAM: 16 ± 5; ABUTA: 18 ± 5, *p* < 0.0001). The ABUTA + I/R group showed a decrease in NO in relation to the I/R group (ABUTA + I/R: 56 ± 10 vs. I/R: 174 ± 13, *p* < 0.0001). The results for the thiols in the renal tissue showed that the ABUTA + I/R group obtained an increase in the thiols in relation to the I/R group (ABUTA + I/R: 17.0 ± 3.0 vs. I/R: 2.0 ± 0.2, *p* < 0.0001).

### 3.7. Histological Analysis

The tubulo-interstitial lesions were evaluated by means of a lesion score ranging from 0 to 4. The I/R group showed an increase in the injury score compared to the SHAM (I/R: 0.41 ± 0.20 vs. SHAM: 0.05 ± 0.02; ABUTA: 0.0 3± 0.01, *p* < 0.0001). The ABUTA + I/R group presented tubulo-interstitial involvement in less than 5% of the focal tissue areas, a value significantly lower than that of the I/R group (ABUTA + I/R: 0.16 ± 0.02 vs. RI: 0.40 ± 0.20, <0.0001), as shown in Figure 5.

Figure 6 shows representative images of the histological analysis of the groups studied. The SHAM (A) and ABUTA (B) groups presented a normal histological section, without tubulo-interstitial alterations, with classification 0 or 0.5 in the tissue. The animals in the I/R (C) group-maintained grade 3 with involvement of 25% to 75% of tubulo-interstitial alterations, presenting tubular atrophy and dilation, in addition to tubulo-interstitial infiltrate. However, animals exposed to I/R and submitted to treatment with *Abuta grandifolia* (D) presented moderate tubular dilation and reduction of tubulo-interstitial infiltrate.

## 4. Discussion

The phytochemicals found in the extract of *Abuta grandifolia* were alkaloid compounds, especially benzyltetrahydroisoquinoline, which is a precursor of several types of skeletons, such as: morphine, aporfinic, bisbenzylisoquinoline, and protoberberinic [39]. Aporfinics constitute one of the largest groups of isoquinoline alkaloids in *Menispermaceae*, presenting a growing list of biological activities of antimicrobial [40] and anti-inflammatory action [38]. 

The EA of *Abuta grandifolia* has also been shown to contain concentrations of flavonoid and oleanolic acid. This compound has actions in modulating biological responses, protecting cells from oxidative damage, stimulating the immune system, and has anti-inflammatory properties [41]. It is suggested that these actions occur through reactions with sulfhydryl groups in proteins [42].

When we analyze acute kidney injury, we find that it is characterized as a rapid decline in the excretory function of the kidneys, deposition of end products of nitrogen metabolism such as urea and creatinine, or decreased urine production. In a rat model for ischemia/reperfusion, early morphological changes (such as tubular dilation and loss of renal microvilli) are observed in all segments of the proximal tubule within hours of the onset of reperfusion [43]. 

The animals belonging to the I/R group, when compared with the animals of the SHAM group, did not demonstrate a significant increase in urine flow. However, there was vasoconstriction and microvascular obstruction, decreased glomerular filtration rate and also decreased renal blood flow, factors considered reliable for the occurrence of iAKI [44,45].

The non-oliguric state can accompany AKI due to prerenal azotemia and a variety of disorders of the renal parenchyma, as well as acute tubular necrosis. Non-oliguric forms of AKI are associated with less morbidity and mortality than oliguric AKI. In addition, renal volume expansion, treatment with agents with diuretic and renal vasodilator effects, can convert oliguric to non-oliguric acute tubular necrosis if administered early in the treatment of AKI [46,47]. The animals in the ABUTA + I/R group, because they received the treatment, produced a significantly greater volume of urine and also showed tubular dilatation, which may explain the weight gain and glomerular hyperfiltration in this group, in addition to the state of improvement in LRA.

It is noteworthy that despite this factor, there was no impact on renal function, since weight variation is not associated with differences in functional maturity of the animals used in the study. It was also found that there is no direct relationship between blood concentration and excretion of renal function markers with renal hyperfiltration, especially when the kidney injury is not severe [48].

However, one characteristic of renal injury is prominent infiltration of leukocytes, especially neutrophils, occurring rapidly after ischemia/reperfusion injury in rodents, which can cause interstitial edema induced by leukocytes, which can increase interstitial pressures and slow down peritubular capillary flow. This activity may represent a change to the extension phases of iAKI and may cause resistance to vasodilator therapies after the onset of AKI [49].

Inhibition of the action of aquaporins (AQP1, AQP2, AQP3) on proximal tubules and distal tubules, together with reduced levels of sodium transporters along the nephron that compromise tubular reabsorption of filtered sodium, increase sodium and water excretion during the recovery phase of ischemia-induced AKI. Occurrence of this action in the first 24 h after ischemia of 30 min bilateral contributes to recovery in moderate iAKI, and urine output may be accentuated with administration of pretreatments for AKI [50,51,52]. Another result of this study, which reproduces the procedures in the same line of investigation, is that the animals in the I/R group presented increased serum creatinine levels, in addition to decreased inulin clearance, confirming the occurrence of tubular injury.

Significant increases in serum creatinine (SCR) levels in ischemic rats and decreased inulin excretion indicated damaged structural integrity of nephrocytes and presence of renal dysfunction. Pretreatment with *Abuta grandifolia* decreased SCR levels and increased inulin clearance, probably by maintaining cell membrane integrity. This result was also verified by histopathological observation, which was used to evaluate the damage to the renal structure, which through pretreatment obtained a significant effect on the preservation of renal cell architecture and decreased infiltration of inflammatory cells [53]. 

In the present study, slight variations in MAP were observed in the groups that received *Abuta*; however, there was no influence on the parameters of renal function. Also with regard to renal hemodynamics, an increase in renal vascular resistance (RVR) was demonstrated in the animals in group I/R, when compared to the SHAM and ABUTA groups.

The striking feature of AKI, considering the role of tubular injury, is a reduction in the glomerular filtration rate (GFR), which in essence implies underlying impairment of hemodynamic regulation. Typically, the response is a sustained increase in renal vascular resistance (RVR) and decreased renal blood flow (RBF) that may be attributable to several factors related to altered intrinsic constriction mechanisms or increased vasoconstrictor production [54].

Impaired renal medullary blood flow may further exacerbate hypoxia in the early stages of reperfusion of ischemia in rodents. In humans, hypoxia is aggravated in grafts by immediate delayed function after transplantation. Sustained hypoxia may inhibit ATP resynthesis, accounting for a higher degree of injury [55].

Ischemia-induced injury can result in damage to both the tubular and microvascular compartments. This may influence the increase in RVR, when added to inflammatory mechanisms, conferring reductions in FSR and GFR with low or no response to vasodilator therapies. However, macrovascular renal blood flow may or may not correlate with glomerular perfusion, because changes in glomerular perfusion can occur even in periods of preserved blood pressure through different specific effects on afferent and efferent arterioles [56]. 

The early vasoconstriction observed in many models impairs blood flow and is probably mediated by several pathways, which, when blocked, produce deleterious effects on the kidney, accompanied by characteristics secondary to inflammation or structural changes in the renal parenchyma that probably sustain reductions in FSR [57]. At this point, hemodynamic therapies may have a more limited role, and therapies should be directed at blocking inflammatory pathways and/or restoring structure [58]. Pretreatment with *Abuta grandifolia* helped in vasodilation of the blood pathways and reduction of RVR, increasing FSR and influencing the adequacy of GFR.

As for the oxidative profile, a non-significant increase in urinary peroxide concentrations and in lipid peroxidation was observed in the ABUTA groups. It must be considered that, naturally, plant organisms have products related to oxidative stress in their constitution, generally antioxidant agents, which are involved with the plant growth and development. These waste products can be absorbed by animal organism through the consumption of plants [59]. In the present study, these increase in concentrations of urinary peroxide and MDA in animals from the ABUTA group was not considered harmful, once no deleterious effect was detected in the renal function studies.

Some of the main causal factors in I/R-induced AKI are based on transient increases in reactive oxygen species (ROS) [57]. There was an increase in urinary peroxides, which are mild and relatively stable oxidants. They are not considered true free radicals, but at pathophysiological relevant concentrations, they can provoke most of the phenotypic changes in endothelial function that are evidenced in post-ischemic (posthypoxic) tissues. They are also involved in the production of many reactive oxygen species and are able to cross the nuclear membrane of the cell and induce damage to the DNA molecule by means of enzymatic reactions. The harmful effects of ROS can be explained by several mechanisms, including cell injury, peroxidation of membrane lipids, denaturation of proteins and, as already mentioned, DNA damage [60]. 

In the I/R group, there was an increase in the parameters of urinary peroxides, which may be one of the reasons for deflagration in the renal tissue of ischemic rats. Treatment with *Abuta grandifolia,* may have contributed to the minimization of the deleterious effects of urinary peroxides and allowed more rapid tissue restructuring in the kidneys of rats in the ABUTA + I/R group. High levels of MDA are indicative of lipid peroxidation, which affects cell membranes under conditions of oxidative stress, because a free radical extracts an electron from a lipid, causing the formation of a fatty acid radical, increasing MDA concentrations and reducing the action of enzymes and antioxidant factors [61]. Our results showed that MDA levels in the body of animals decreased to a greater extent in rats receiving doses of *Abuta grandifolia*, compared to rats in the I/R group. 

NO is synthesized in the endothelium by Ca^2+^ calmodulin-dependent nitric oxide synthase (eNOS) and can regulate vascular tone as well as attenuate smooth muscle cell proliferation [62]. A significant increase in NO in the I/R group has been shown to cause excessive generation of free radicals; the actions of NO are modulated by its rapid reaction as a superoxide radical [63]. This reaction is known to produce unusual and reactive peroxide, that is, peroxynitrite, which represents the fusion of oxygen and NO radicals, further complicating oxidative injury [64]. Thus, the reduction of NO by the extract of *Abuta grandifolia* is extremely significant, since it advances in the antioxidant component by reducing peroxides and avoids, through the reduction of NO, additional oxidative damage.

In the present study, the amounts of renal thiols were significantly reduced in rats in the I/R group, where in the presence of AKI, the action of antioxidant enzymes such as transferase (GST), glutathione peroxidase, and superoxide dismutase (SOD) is decreased in the presence of oxidative injury. This also causes a positive change towards pro-oxidants and induces a decrease in the levels of non-enzymatic antioxidants, such as reduced glutathione (GSH), protein, and non-protein thiols [65]. 

The extract of *Abuta grandifolia* improved the parameters of total thiols, conferring an antioxidant property. In many ways, antioxidants perform their functions against nephrotoxic agents, including maintaining intracellular GSH concentrations, restoring cellular defense mechanisms, blocking lipid peroxidation, and protecting kidney cells [64,65].

The activity of antioxidant enzymes often depends on the participation of enzyme cofactors, especially antioxidants of dietary origin. Considering the metabolites found in *Abuta grandifolia* extract, the results of Oliveira’s (2023) review [11] allowed inferring that the antioxidant action of the plant is closely related to the concentration of alkaloids and polyphenols found in the bark extracts. Aware of protein oxidation in renal I/R injury and having established the vital role of ROS in the pathogenesis of I/R injury, we hypothesized that the donor hydrogens in the alkaloids and phenolic compounds may have reacted and neutralized the ROS, this process being considered more plausible to characterize the antioxidant action of *Abuta grandifolia* [59,66]. 

Tubular dysfunction is the main consequence related to the maintenance of AKI damage by I/R, which induces cell death in the epithelium, tubular fibrosis and renal dysfunction. Tubular cells are vulnerable to ischemia, and the response is variable and multifactorial, depending on the time of hypoxia, the metabolic needs of the tissue, the contribution of collateral circulation, and humoral factors [67]. 

We investigated a series of inflammatory and vasoconstrictor events that occur in the renal tissue in hypoxia and after the formation of ROS, concomitant with reperfusion activate hemodynamic mechanisms, especially those activated by the juxtaglomerular apparatus in the dense macula, activating the routes of action of the renin-angiotensin-aldosterone system. This causes glomerular damage associated with damage to the Bowmann capsule; depletion in podocytes; and inflammatory, pro-oxidant, ischemic and fibrotic responses. This leads to mesangial matrix accumulation, glomerular basement membrane thickening, and endothelial dysfunction [68,69].

In this study, photomicrographs of kidney staining revealed thickening of the basement membrane of the glomeruli and renal tubules with loss of brush border in the tubular epithelium as the main pathological lesions in the I/R group. The severity of the lesions observed in the various groups was confirmed by the lesion scores. There were significantly higher levels of damage in endothelial, glomerular, tubular and interstitial lesions in animals in the I/R group, compared to animals in the ABUTA + I/R group, which did not present significant lesions. Our findings indicate that the administration of *Abuta grandifolia* prevented the progression of the extension area with tissue injury after I/R.

The results point to the involvement of cellular respiration and redox balance in the processes of death and recovery, as well as the mechanism of action of the substance under study. It is known that the kidney has the ability to restore itself after an ischemic insult, because the unaffected cells develop mechanisms to recover the functional integrity of the nephron. We highlight that there is a dynamic relationship between renal tissue repair and the progression or regression of ischemic AKI. Early tissue repair stops the progression of the lesion and recovers the kidney [69].

Another relevant aspect is the antioxidant potential. The existing metabolites in *Abuta grandifolia* attenuate the oxidative damage of cells, indicating activation of antioxidant signaling pathways in a time- and dose-dependent manner, modulating inflammation and apoptosis induced by I/R. So some molecular mechanism directed by such secondary compounds causes a decrease in the action of enzymes and pro-apoptotic pathways, already evidenced by inhibition of oxidative stress, adequacy of renal function, and restructuring of endothelial cells in renal tissue.

The present findings show that the oral administration of *Abuta grandifolia* produces protective effects for renal function and hemodynamic parameters, preventing lipid peroxidation, in addition to inhibiting the oxidation of proteins via inactivation of ROS. It is suggested that the protective effect of the plant against tissue and metabolic deflagrations in renal reperfusion ischemia may be related to the antioxidant and anti-inflammatory activities of substances present in its extract, such as flavonoids and alkaloids, which have significant nephroprotective and diuretic properties.

## 5. Conclusions

Treatment with *Abuta grandifolia* promoted functional renoprotection and confirmed its antioxidant activity by reducing oxidative metabolites and increasing thiols, which reduced tubulointerstitial injury. Further investigations are needed to explore the mechanism of action of the extract against iAKI.

## Figures and Tables

**Figure 1 cells-12-01688-f001:**
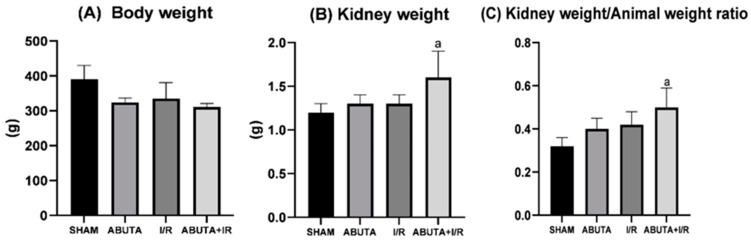
Physiological parameters. (**A**) Body weight showed no statistically significant differences between the groups. (**B**) Kidney weight showed a statistically significant increase between the ABUTA + I/R group compared to the SHAM, ABUTA and I/R groups, [F-4.323], *p* = [*p* < 0.0001]. (**C**) Kidney/body weight ratio demonstrated a statistically significant increase between the ABUTA + I/R group compared to the SHAM, ABUTA and I/R groups, [F-6.92], *p* = [*p* < 0.0001]. *n* = 5 per group. Statistical analysis by one-way ANOVA. Values represent the mean ± standard deviation. a: SHAM group.

**Figure 2 cells-12-01688-f002:**
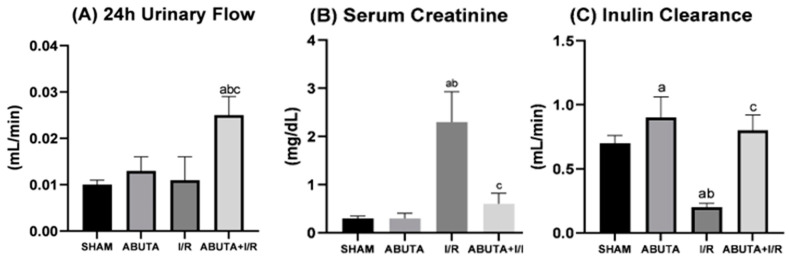
Renal function. (**A**) 24-h urinary flow showed a statistically significant increase in the ABUTA + I/R group compared to the SHAM, ABUTA and I/R groups, [F-18.92], *p* = [*p <* 0.0001]. (**B**) Serum creatinine showed a statistically significant decrease in the ABUTA + I/R group compared to the I/R group, [F-31.45], *p* = [*p* < 0.0001]. (**C**) Inulin clearance showed a statistically significant increase among the ABUTA + I/R group compared to the I/R group, [F-35.36], *p* = [*p* < 0.0001]. *n* = 5 per group. Statistical analysis by one-way ANOVA. Values represent the mean ± standard deviation. a: SHAM group; b: ABUTA group; c: Ischemia/Reperfusion Group.

**Figure 3 cells-12-01688-f003:**
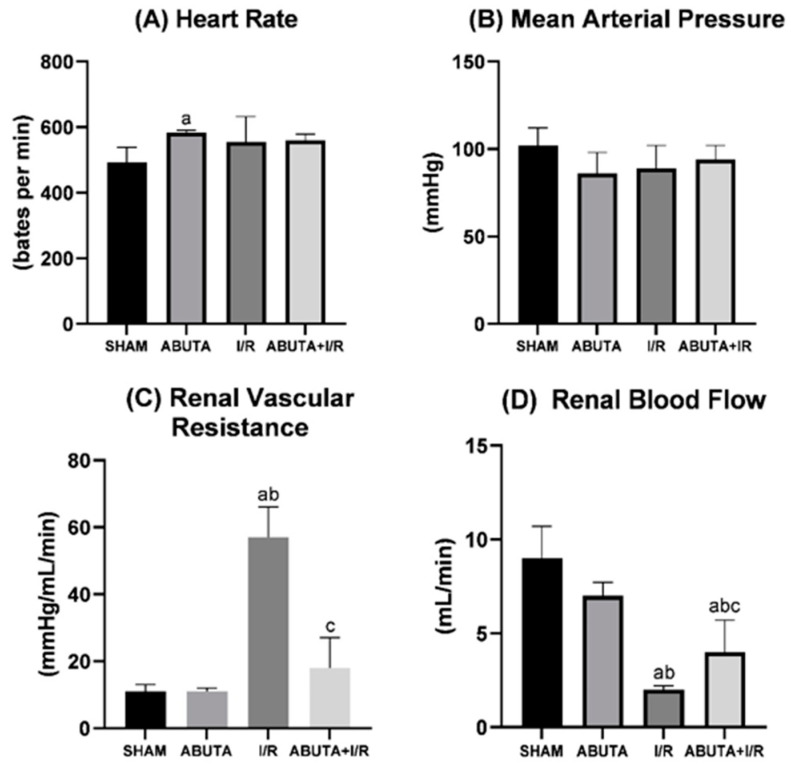
Renal hemodynamics. (**A**) Heart rate showed a statistically significant increase in the ABUTA group compared to the SHAM, I/R and ABUTA + IR groups, [F-3.628], *p* = [*p* < 0.0001]. (**B**) Mean arterial pressure showed no statistically significant differences between groups, [F-6.89], *p* = [*p* < 0.0001]. (**C**) Renal vascular resistance showed a statistically significant increase in the I/R group compared to the SHAM, ABUTA and ABUTA + IR groups, [F-62.13], *p* = [*p* < 0.0001]. (**D**) Renal blood flow demonstrated a statistically significant increase between the ABUTA + I/R group compared to the I/R group, [F-32.54], *p* = [*p* < 0.0001]. Statistical analysis by one-way ANOVA. *n* = 5 per group. Values represent mean ± standard deviation. a: SHAM group; b: ABUTA group; c: Ischemia/Reperfusion Group.

**Figure 4 cells-12-01688-f004:**
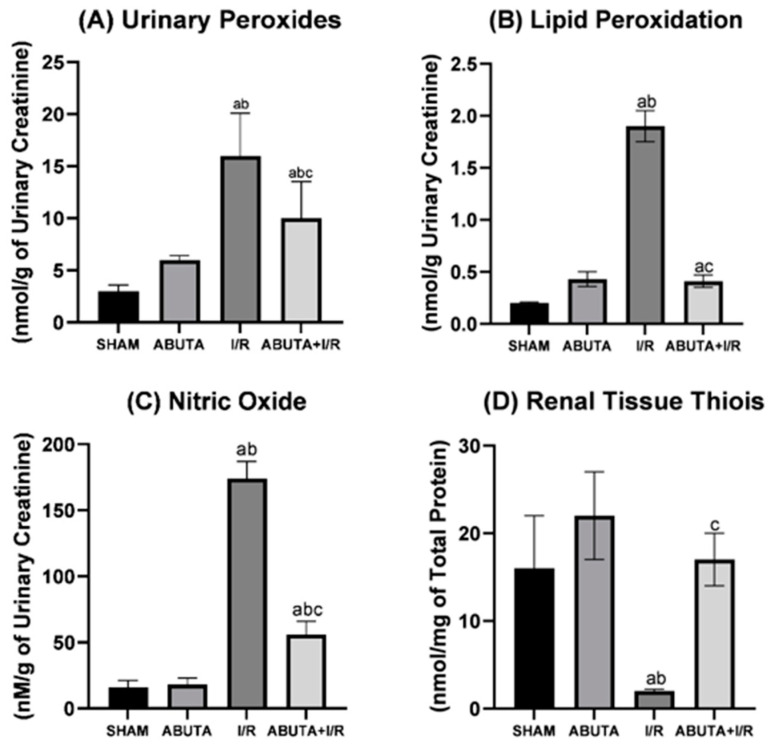
Oxidative profile. (**A**) Urinary peroxides demonstrated a statistically significant decrease in the ABUTA + I/R group compared to the SHAM, ABUTA and I/R groups, [F-16.38], *p* = [*p* < 0.0001]. (**B**) Lipid peroxidation showed a statistically significant decrease in the ABUTA + I/R group compared to the SHAM and I/R groups, [F-344], *p* = [*p* < 0.0001]. (**C**) Nitric oxide showed a statistically significant decrease in the ABUTA + I/R group compared to the I/R group, [F-274.2], *p* = [*p* < 0.0001]. (**D**) Renal tissue thiols showed a statistically significant increase in the ABUTA + I/R group compared to the I/R group. [F-21.01], *p* = [*p* < 0.0001]. *n* = 5 per group. Statistical analysis by one-way ANOVA. Values represent the mean ± standard deviation. a: SHAM group; b: ABUTA group; c: Ischemia/Reperfusion Group.

**Figure 5 cells-12-01688-f005:**
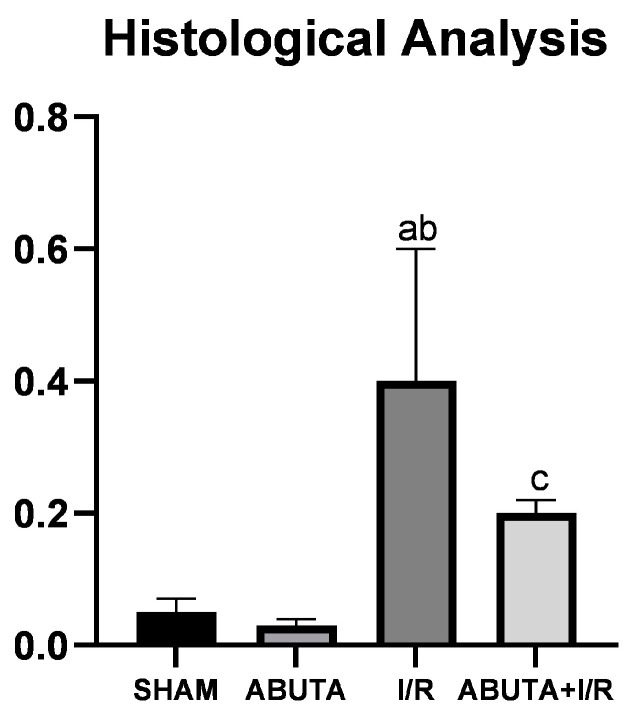
Renal histological analysis showed a statistically significant decrease in the lesions in the renal tissue in the ABUTA + I/R group, in relation to the I/R group, [F-117.3], *p* = [*p* < 0.0001]. *n* = 5 per group. Statistical analysis by one way ANOVA. Values represent mean ± standard deviation. a: SHAM group; b: ABUTA group; c: Ischemia/Reperfusion group.

**Figure 6 cells-12-01688-f006:**
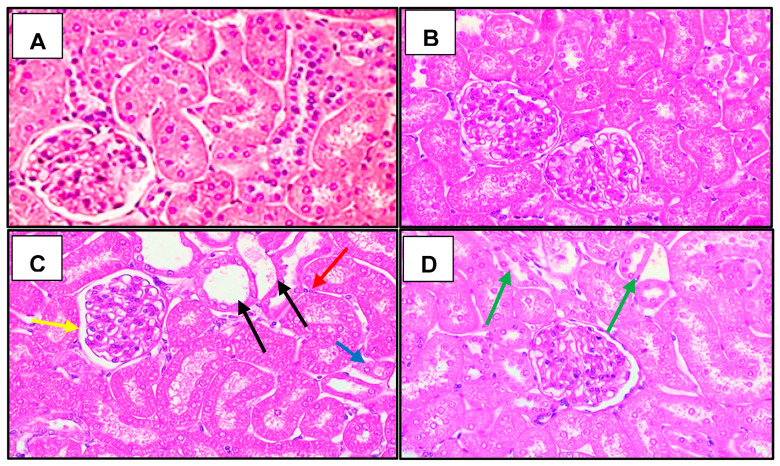
Renal Histology: necrosis, cellular inflammatory infiltrate, tubular dilation and atrophy. (magnification ×400) of the groups: SHAM (**A**), ABUTA (**B**), I/R (**C**), ABUTA + I/R (**D**). Black arrows: necrosis; Yellow arrow: Bowman’s space dilation; Red arrow: Infiltration of inflammatory cells; Blue arrow: tubular atrophy; Green arrows: tubular regeneration.

**Table 1 cells-12-01688-t001:** Antioxidant activity of *Abuta grandifolia*.

Sample	% Antioxidant	
	DPPH (517 nm)	ABTS (734 nm)
Aqueous Extract	76.03%	89.55%

## Data Availability

The data presented in this study are available on request from the corresponding author.

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
