# Peer review of "Amazonia Phytotherapy Reduces Ischemia and Reperfusion Injury in the Kidneys"

_cells, 2023, doi:10.3390/cells12131688_

Round 1
Reviewer 1 Report
The manuscript submitted (cells-2299373) by Ferreira de Oliveira was aimed to evaluate the effect of the Amazonian plant Abuta grandifolia on renal function in rats with acute renal injury following I/R. The authors conclude that Abuta grandifolia has renoprotective effects, through its antioxidant activity.
From the conceptual point of view, the work is appropriate; however, it has some methodological shortcomings that should be clarified.
· According to the journal's instructions, and the existing template , the Material and Methods section should follow the introduction section, not the discussion.
· As usual, and as stated in the journal's instructions, the title of the figures should be below them, not above them.
· The authors must detail the age of the animals used.
· Regarding the statistical section, how was the sample calculation method used to determine the size of each experimental group? Why did you choose experimental groups of five animals?
· Again, regarding the statistical analysis of the manuscript, there is no mention in the material and methods section indicating how the statistical analysis was performed.
· No postoperative analgesia was applied, provide a reason for not applying any type of postoperative analgesia.
· According to the assessment of glomerular filtration rate, the authors indicate that they administered inulin through the jugular vein, first as a bolus and then by continuous infusion for 2 hours. During this infusion period, were the animals under sedation or kept awake?
Reviewer 2 Report
The manuscript: “Amazonia phytotherapic reduces injury by ischemia reperfusion in the kidney” by Oliveira et al. In the present study, the authors demonstrate that the plant extract Abuta grandifolia promotes renoprotection by reducing oxidative stress and morphological changes after renal ischemia and reperfusion. Although an interesting topic, there exist a number of concerns about the paper as presented. The comments are offered as followed:
Introduction
1. In the introduction, the authors cite a reference from 2008 that did not study the analyzed species to justify that “there is still no scientific basis on the effects of these substances on renal function”.
Results
1. The authors describe that Similarly, it was observed a significant increase in the body in the ABUTA+I/R versus SHAM, ABUTA and I/R. However, in figure 1 there was a significant difference in the kidney weight and the kidney weight/animal weight ratio but there was no difference in the body weight.
2. Correct in the legend of figure 1 and in the figure “Body” weight/Animal weight ratio.
3. What is the reduction in FSR shown on line 162?
4. In figure 4 in (B) lipidic peroxidation does the ABUTA+I/R versus ABUTA really show a significant difference?
5. In figure 5 the statistical differences are not indicated.
6. In Figure 6, the authors must indicate the presence of necrosis, cellular inflammatory infiltrate, tubular dilation or atrophy.
7. Figure legend should be more detailed.
Discussion
1. In the discussion the authors say that in the ischemia/reperfusion rat model tubular occurs dilation and loss of renal microvilli in all segments of the proximal tubule, and that tubular dilation may account for the increased weight and glomerular hyperfiltration. However, in the I/R group this did not happen. How can the authors justify this?
2. The authors use the abbreviation GFR, what does this abbreviation mean?
3. In an ischemia/reperfusion rat model, early morphological changes (such as tubular dilation and loss of renal microvilli) are observed in all segments of the proximal tubule within hours of the onset of reperfusion, such tubular dilation may account for the increased of weight and glomerular hyperfiltration in the ABUTA+I/R group. And in the I/R group what happens?
4. The discussion needs to be improved. The authors discuss the parameters analyzed without relating them to the results obtained mainly in the I/R group.
Methods
1. The authors describe the ABUTA group: “animals that received Abuta grandifolia (400mg/kg, 1x a day, orally - PO, 5 days)”; Have not these animals undergone surgery? If so, this information should be added to the abstract and methodology.
2. And did the SHAM and I/R groups receive the vehicle?
3. Without the description of the score and the reference 83, it was not possible to evaluate the renal histology analyses.
4. How did the authors assess tubular dilation or atrophy.
References
1. Some quotes may be deleted.
2. In the methodology of renal histology, the authors cite reference 83.
However, it is not found in the list of references.
Round 2
Reviewer 1 Report
the authors have responded adequately to the questions posxed.
Author Response
The present manuscript has been revised in English to make the necessary corrections, and corrections suggested by the editor have also been made and are attached.
